# Cross-Sectional Study of University Students’ Attitudes to ‘On Campus’ Delivery of COVID-19, MenACWY and MMR Vaccines and Future-Proofing Vaccine Roll-Out Strategies

**DOI:** 10.3390/vaccines10081287

**Published:** 2022-08-10

**Authors:** Adam Webb, Mayuri Gogoi, Sarah Weidman, Katherine Woolf, Maria Zavala, Shamez N. Ladhani, Manish Pareek, Lieve Gies, Christopher D. Bayliss

**Affiliations:** 1Department of Genetics and Genome Biology, University of Leicester, Leicester LE1 7RH, UK; 2Department of Respiratory Sciences, University of Leicester, Leicester LE1 9HN, UK; 3Faculty of Medicine, University College London Medical School, London WC1E 6DE, UK; 4Immunisation and Countermeasures Division, Public Health England Colindale, London NW9 5EQ, UK; 5Department of Infection and HIV Medicine, University Hospitals of Leicester NHS Trust, Leicester LE1 5WW, UK; 6School of Media, Communication and Sociology, University of Leicester, Leicester LE1 7RH, UK

**Keywords:** COVD-19, meningitis, MMR, vaccine hesitancy, university students, vaccine uptake

## Abstract

University students are a critical group for vaccination programmes against COVID-19, meningococcal disease (MenACWY) and measles, mumps and rubella (MMR). We aimed to evaluate risk factors for vaccine hesitancy and views about on-campus vaccine delivery among university students. Data were obtained through a cross-sectional anonymous online questionnaire study of undergraduate students in June 2021 and analysed by univariate and multivariate tests to detect associations. Complete data were obtained from 827 participants (7.6% response-rate). Self-reporting of COVID-19 vaccine status indicated uptake by two-thirds (64%; 527/827), willing for 23% (194/827), refusal by 5% (40/827) and uncertain results for 8% (66/827). Hesitancy for COVID-19 vaccines was 5% (40/761). COVID-19 vaccine hesitancy was associated with Black ethnicity (aOR, 7.01, 95% CI, 1.8–27.3) and concerns about vaccine side-effects (aOR, 1.72; 95% CI, 1.23–2.39). Uncertainty about vaccine status was frequently observed for MMR (11%) and MenACWY (26%) vaccines. Campus-associated COVID-19 vaccine campaigns were favoured by UK-based students (definitely, 45%; somewhat, 16%) and UK-based international students (definitely, 62%; somewhat, 12%). Limitations of this study were use of use of a cross-sectional approach, self-selection of the response cohort, slight biases in the demographics and a strict definition of vaccine hesitancy. Vaccine hesitancy and uncertainty about vaccine status are concerns for effective vaccine programmes. Extending capabilities of digital platforms for accessing vaccine information and sector-wide implementation of on-campus vaccine delivery are strategies for improving vaccine uptake among students. Future studies of vaccine hesitancy among students should aim to extend our observations to student populations in a wider range of university settings and with broader definitions of vaccine hesitancy.

## 1. Introduction

Young people are an important risk group for vaccination programmes due to their high mobility, inexperience of accessing medical systems and relatively high levels of vaccine hesitancy compared to older populations [1,2]. Within this group, university students are particularly at risk of contracting and transmitting infectious diseases because of their high levels of transmission-associated behaviours and mixing of geographically diverse intakes [3,4,5]. These risks have been exemplified by outbreaks of COVID-19 on university campuses as students returned to campus-based activities after initial lockdowns [6,7]. Facilitating access of university students to vaccines is a key mechanism for enhancing vaccine uptake and preventing infectious disease outbreaks while minimising the need for highly restrictive measures such as lockdowns, social distancing and online learning.

Multiple previous studies have been conducted to determine the reasons for COVID-19 vaccine hesitancy in the general population. Vaccine hesitancy or acceptance has been assessed by a range of measures including use of the SAGE guidelines, Likert-scaled acceptance questions, attitudinal measures and actual uptake or intention to uptake (as utilised herein) [8,9,10,11]. A recent meta-analysis of vaccine acceptance in higher income countries reported vaccine hesitancy rates of at least 30% in half the studies (*n* = 97) with lower socioeconomic status being the most impactful contributory factor in lower-middle income countries/regions and perceived vaccine safety in more affluent countries/regions [8]. Common demographics for COVID-19 vaccine hesitancy in the reported literature include females, younger age groups, being from a minority ethnic group and lower education or income levels [12,13,14]. Studies of student populations have yielded a range of findings. Factors associated with higher vaccine acceptance are knowledge of COVID-19 vaccines, trust in authorities and high perceived vaccine safety and effectiveness and uptake of vaccination in the family [10,11,15,16,17,18,19,20]. Consequently, perceived accessibility barriers (physical or financial), concerns about vaccine side-effects, speed of development and previous COVID-19 infection have been associated with vaccine hesitancy [11,18,20,21,22,23]. A potentially important issue is whether news about the rare but serious side-effect of blood clots associated with the licensed AstraZeneca COVID-19 vaccine might have affected uptake among students [24,25].

Prior to the COVID-19 pandemic, a major concern for student populations was the prevention of cases and outbreaks of meningococcal disease, measles and mumps. Rising levels of infections due to a MenW cc11-lineage strain led to introduction of the MenACWY vaccine into the UK school-age vaccination programme and new university entrants from August 2015 [26,27]. Outbreaks of mumps among students were also observed in 2019, leading to student-focussed information campaigns to encourage uptake of the MMR vaccine [28]. The MenACWY and MMR vaccines are currently offered free of charge to all university students in the UK, including overseas students. National lockdowns to contain the rapid spread of SARS-CoV-2 in 2020 led to major reductions in cases of meningococcal disease, measles and mumps, but there is now a concern that ending lockdowns and increased social mixing could lead to rises in these serious vaccine-preventable diseases [29,30]. These effects may be compounded by disruption of school-based immunisation programmes during the pandemic, which may have resulted in a serious risk of long-term weakening of individual and herd (population) protection.

Studies prior to the COVID-19 pandemic reported uptake rates of the MenACWY vaccines among students at 68–71% [9,27,31]. In general, students are expected to obtain their vaccines prior to arrival at university. However, uptake can be enhanced by ‘on campus’ vaccine campaigns as exemplified by the University of Nottingham’s highly effective delivery of the MenACWY vaccine for incoming university entrants [27]. Vaccine hesitancy has been examined for the MenACWY vaccine. Blagden et al. [9] reported that vaccine uptake was strongly associated with a high perceived effectiveness of the vaccine but did not find any barriers, such as vaccine side-effects or inconvenience. A meta-study by Wishnant et al. [32] found that the only factors strongly associated with uptake of meningococcal vaccines among students were perceived risks of contracting meningococcal disease and the severity of meningococcal infections. Overall, these studies indicate that vaccine hesitancy is not a major barrier to meningococcal vaccine uptake but are equivocal about how vaccine uptake can be increased.

To evaluate the barriers to uptake of vaccines among students and to inform university vaccination policies, we assessed the attitudes, knowledge, perceived vaccine status and willingness for uptake of COVID-19, MMR and MenACWY vaccines among university students during the roll-out of COVID-19 vaccines to 18-year-olds and above in the UK.

## 2. Materials and Methods

### 2.1. Ethics

All study participants gave their informed consent for inclusion before participation in the study. The study was conducted in accordance with the Helsinki Declaration and ethical approval was given by the University of Leicester (UoL) Medicine and Biological Sciences Research Ethics Committee (reference number 29522).

### 2.2. Context of Questionnaire Delivery and Derivation

The questionnaire was emailed to students on three occasions between the 1st and 21st of June 2021. Access to COVID-19 vaccines in the UK was extended to the 25–29, 23–24, 21–22 and 18–20 age brackets on the 7th, 15th, 16th and 17th June 2021, respectively (https://www.england.nhs.uk/2021/06/21-and-22-year-olds-to-be-offered-covid-19-jab-from-today/, accessed on 9 August 2022 and https://www.england.nhs.uk/2021/06/nhs-invites-all-adults-to-get-a-covid-jab-in-final-push/, accessed on 9 August 2022). Prior to these dates, only healthcare workers and medical students as well as individuals in vulnerable categories were eligible for COVID-19 vaccines in the <30-year-old age bracket. At the time the questionnaire was designed, we assumed most students would be unvaccinated when they completed the questionnaire.

### 2.3. Questionnaire Delivery, Structure and Content

The questionnaire was administered via Online Surveys. Between 1 and 21 June 2021, the research team sent an invitation email, and two reminder emails, to all 10,869 campus-based University of Leicester undergraduate students (Appendix A). Each invitation email contained a unique link to the questionnaire that could not be reused. Completed questionnaires were de-identified by automatic assignment of another unique identifier by the software, thereby uncoupling the questionnaires from the original email address. The initial email included an invitation to voluntarily participate in follow-up interviews and a prize draw with five prizes of GBP 200 being offered and subsequently delivered.

The questionnaire consisted of a participant information sheet followed by three informed consent questions. Remaining parts of the questionnaire were only accessible if approval to all three consent questions was granted. Apart from 28, questions were compulsory; ‘Prefer not to answer’ or ‘Don’t know’ responses were included for most questions so that participants could opt not to provide specific responses (Appendix A). The questionnaire consisted of 29 questions split into four sections: demographics, vaccines, experiences of COVID-19 disease and other pandemic experiences (e.g., harassment). Questions were multiple choice or scaled answers with one free text box (Question 28) and three questions with answer-dependent questions (Appendix A). Questions 2, 6, 7, 10, 12, 17, 19, 20 and 26 were identical to or modifications of questions utilised in UK-REACH questionnaire 2_ver_1.2 (23 March 2021). Questions 13–16 were written by the authors and piloted with University of Leicester students prior to the pandemic as part of another study [31]. Questions 3–5, 8, 9, 11, 21–25, 27–29 were written by the authors for this study. Question 26 is the self-determination scale. Question 17 utilised four statements from the VAX scale of Martin and Petrie [33]. A VAX score was derived for each participant by reversing the scores for statements 17.2, 17.3 and 17.5 followed by rescaling the sum of all four scores on a 0 to 1 scale that represents maximum to minimum hesitancy, respectively. Internal consistency for VAX score was tested with Cronbach’s alpha.

### 2.4. Definitions of Primary Endpoints

Primary endpoints in our analyses were vaccine hesitancy, VAX scores and willingness for on-campus vaccination programmes. Vaccine hesitancy was defined as providing a response to question 10 that included the phrase ‘have decided not to have the vaccine’. Vaccine-willing students were those whose response included either ‘I have already had’ or ‘intend to have the vaccine’. VAX scores have been utilised as predictors of vaccine hesitancy [33]. VAX scores were derived for all students and analysed for differences between ethnic groups and term time residence locations as an alternate measure of vaccine hesitancy. The potential utilisation of on-campus vaccination programmes was defined based on responses to question 15 split between those in favour (definitely increase, somewhat increase) and those who were ambivalent (neither, somewhat decrease, definitely decrease).

### 2.5. Statistical Analysis

Sample size was determined by the total number of surveys completed with all responses being considered in the analysis. De-identified survey responses were analysed using R version 4.0.3 with the tidyverse (data handling), jsonlite 1.7.1 (data extraction), ggplot2 3.32 (general graphing), gtsummary 1.4.2 (tabulation), UpSetR 1.4.0 (graphing of sets) and likert 1.3.5 (graphing of Likert-style responses) packages [34,35,36,37,38,39,40] all downloaded via the Comprehensive R Network (CRAN; https://cran.r-project.org). In order to determine if the demographics of our study participants were similar to those of other UK universities and for weighting of the multivariable analyses, we obtained demographic data from HESA (Higher Education Statistics Agency, Cheltenham, UK; https://hesa.ac.uk, accessed on 2 August 2021). Similarly, we compared vaccination rates in our study with local, regional and national vaccination rates obtained from the UK Coronavirus Dashboard (https://coronavirus.data.gov.uk, accessed on 6 July 2021).

Univariable analyses were performed on unweighted survey results using chi-squared, Fisher’s exact or Wilcoxon’s rank sum tests. False discovery rate (FDR) correction was used to adjust *p*-values for multiple testing. COVID-19 vaccine hesitancy and preference for on-campus vaccinations (COVID-19 and MMR) were dichotomised and used as dependent variables. 

Multivariable analysis was performed using logistic regression (glm; binomial link function) on both unweighted and weighted survey results, with vaccine hesitancy and preference for on-campus vaccinations (COVID-19 and MMR) as dependent variables. Predictors included gender, ethnic group, age group, course studied, year of study, experience of harassment, experience of COVID-19-related death, concern over side-effects from the Oxford/AstraZeneca vaccine, concern over hospitalisation from COVID-19, concern over spreading COVID-19, home area (local, national, international), residence while studying (home, halls of residence, private accommodation, other) and a psychometric score on self-determination/fatalism. Home area was derived using information on postcodes and international status with students being classified as local if they came from either Leicester or the wider county of Leicestershire, ‘national’ for students from the rest of the UK and ‘international’ for students ordinarily living overseas. Experiences of COVID-19-related deaths were classified into a Yes or No category according to responses to question 21 (Appendix A) with the Yes responses including family members, friends or others. Survey data were weighted using the raking method in the survey package version 4.0 [41] for R based on national student distributions for ethnic group (White, Asian, Black and other) and gender (Appendix A). Constraints in the national data made it necessary to remove students of unknown gender (*n* = 6). Given the low rate of vaccine hesitancy and the inherent limitations on creating meaningful and representative training and test data subsets, no attempt was made to assess model performance (e.g., using ROC AUC). Multivariate models were used to identify significant factors influencing vaccine hesitancy rather than for the purpose of generating an effective predictive model of hesitancy.

Statistical differences between the distributions of VAX scores for different groups were determined using pairwise Dunn tests with FDR correction.

All tests were two-sided with a corrected *p*-value (FDR correction) of <0.05 considered significant.

## 3. Results

### 3.1. Response Rate, Sample Characteristics and COVID-19 Vaccination Uptake

In June 2021, all University of Leicester (UoL) undergraduate students were invited to participate in a study of the uptake and attitudes to COVID-19 vaccines. Complete answers were provided by 7.6% (827/10,869) of participants. Respondents were young (94% 18–25-year-olds), ethnically diverse (25% Asian, 8% Black, 58% White, 9% other) and included 10% (*n* = 86) international students. Response rates were higher among females (11% above the level for UK universities and 14% above the UoL level) and had an ethnicity profile intermediate between UoL and UK university undergraduate populations (Appendix A). The distribution among year of study was, however, strongly representative of the UoL population (Appendix A). Two thirds (64%) of students (527/827) reported having had a COVID-19 vaccine at the time of questionnaire completion: 74% (390/527) had Pfizer/BioNTech, 23% (121/527) AstraZeneca and 3% (16/527) another vaccine. A further 194 students (23%) expressed a willingness to become vaccinated, giving a total of 85% who had been or were willing to be vaccinated. Results for 66 students were excluded from further analysis of vaccine hesitancy due to uncertainty about their intention to vaccinate (the selected response was ‘I have not had a vaccination but have been told that I will be offered a vaccination in the near future’). Removing these students from the denominator gave an overall willingness rate of 95% (721/761). There were 40 students (5%) who indicated that they had refused or would refuse a COVID-19 vaccine.

### 3.2. Univariable Analysis of Vaccine Hesitancy

The results of the univariable analysis for vaccine hesitancy are shown using 40 students (5%) who indicated that they had refused or would refuse a COVID-19 vaccine.

Ethnicity, concerns around side-effects (particularly the AstraZeneca vaccine), concerns around spreading COVID-19 to others, place of residence while studying and VAX score were all found to be significantly associated with hesitancy after correcting for multiple testing. There was a weak trend for an association of age with vaccine hesitancy; this could not, however, be explored further due to banded collection of age data and the narrow age range of this cohort.

Analysis of the individual VAX scale questions (see Appendix A) showed that only 29% of hesitant students disagreed that natural exposure to a disease was safer than vaccination compared to 80% of vaccine-willing students. By contrast, 70% of hesitant students, but also 54% of willing students, had concerns about the safety of vaccines (the statement was ‘Although most vaccines appear to be safe, there may be problems that we have not yet discovered’). Approximately half (49%) the hesitant students and 82% of willing students agreed that >95% vaccine coverage was required to prevent the spread of infectious diseases.

A surprising observation was that high proportions of both vaccine-willing (42%, 299/721) and vaccine-hesitant (48%, 19/40) students had experienced a COVID-19 death among relatives or other acquaintances (Appendix A). This outcome was, however, not associated with differences in hesitancy (Table 1).

### 3.3. Multivariable Analysis of Vaccine Hesitancy

The multivariable analysis identified associations between vaccine hesitancy and ethnicity, course of study, side-effects and place of term-time residence, as found in the univariate analysis, and additionally with experiences of death among contacts (Table 2). For course studied, those studying medicine and allied professions (e.g., midwifery, nursing and physiotherapy) had a significantly lower likelihood of being vaccine-hesitant (OR 0.1, 95% CI 0.02–0.5, adjusted *p* = 0.021) compared to humanities, law and social sciences. For ethnicity, hesitancy among Black students had a high odds ratio (OR 7.01, 95% CI 1.81–27.3, adjusted *p*-value = 0.021) as compared to White students (Table 2). Students living in private accommodation (OR 0.13, 95% CI 0.04–0.38, adjusted *p* = 0.004) were less vaccine-hesitant than students living at home.

Hesitancy was strongly associated with concerns over side-effects from the AstraZeneca (Table 2) and Pfizer/BioNTech vaccines (OR 2.1, 95% CI 1.5–3.0, adjusted *p* < 0.001; data not shown). Concerns about side-effects were, however, lower for the Pfizer/BioNTech vaccine (Appendix A). Surprisingly, the multivariate analysis detected a positive association between experiences of a COVID-19-related death in a family member, friend or other contact with vaccine hesitancy (Table 2). This association remained even when only close contacts (friends; family) were considered (odds ratio 6.4, 95% CI 1.9–21.6, adjusted *p* = 0.02; data not shown).

### 3.4. Analysis of VAX Scores

The Asian ethnic group had significantly lower VAX scores than both White and other ethnic groups, while the Black ethnic group had significantly lower VAX scores than White ethnicity, indicating a higher level of vaccine hesitancy in Asian and Black groups (Figure 1). Similarly, we observed that home students had significantly lower VAX scores than students living in private or other accommodation (Figure 1). The mean VAX score for students living in halls was higher than but not significantly different from those living at home, indicating a trend for home students to be more vaccine-hesitant than those who lived in other locations during this academic year. VAX scores in our sample had a low but acceptable internal consistency score (Cronbach’s alpha = 0.62; 95% CI 0.58–0.66) and a negative association with our independent measure of vaccine hesitancy (rank biserial correlation *r* = 0.63; Wilcoxon rank sum test *p* < 0.0001), as expected. 

### 3.5. Knowledge of MMR and MenACWY Vaccine Status among Students

Views on MMR and MenACWY vaccines are shown in Table 3. Very few UK students (2–4%) self-reported not having had the MMR or MenACWY vaccines, but an additional 8% did not know if they had had their MMR vaccine and 23% did not know if they had had their MenACWY vaccine (Table 3). International students were more likely not to know their vaccination status compared to local students (Table 3). Additionally, 15% of UK students did not know that the MMR and MenACWY were available free of charge in the UK and 6% reported not knowing that COVID-19 vaccines were also available free of charge. Again, these proportions were significantly higher among international students. More than half (57% and 61%, respectively) of students favoured on-campus MMR/MenACWY and COVID-19 vaccine provision, respectively. UK-based international students were also highly supportive of this provision with 52–62% selecting a ‘definitely increase’ response for these vaccines (Table 3).

### 3.6. Univariable and Multivariable Analyses of Attitudes to On-Campus Vaccinations

The univariable analysis of on-campus MMR/MenACWY vaccine programmes identified a significant association with MMR vaccine status indicating that those who responded with either a ‘Yes’ or ‘Don’t Know’ response for their vaccine status were in favour of these programmes (Appendix A). However, these responses were not significant in the multivariate analysis after correction for multiple testing (Appendix A). The univariable analysis of on-campus COVID-19 vaccine provision found significant associations with COVID-19 vaccine hesitancy and term-time residence (Appendix A). In the multivariable analysis, vaccine hesitancy was negatively correlated with on-campus provision (Appendix A). Multivariable logistic regression of term-time residence indicated that students studying in halls (OR 3.5 95% CI 1.6–7.6, adjusted *p* = 0.021) or private accommodation (OR 2.6, 95% CI 1.3–4.99, adjusted *p* = 0.03) were in favour of this provision.

## 4. Discussion

University students are a critical group for illness and spread of infectious diseases and hence are an important target for vaccination programmes. Our survey of University of Leicester students was unique in that we evaluated both attitudes to and mechanisms for uptake of the three major vaccines targeted to this population group in the UK. Our study indicates that ethnicity, concerns over side-effects and place of residence are key determinants of COVID-19 vaccine hesitancy. We also found high levels of uncertainty among students about their MenACWY and MMR vaccine status. As an approach to facilitating vaccine uptake, students were asked about on-campus provision of vaccines and reported being in favour of this approach. Based on our findings we elaborate key recommendations for improved vaccine delivery to this population sector.

Our study observed a high level of uptake (64%) despite this age group only becoming eligible for COVID-19 vaccines during the data collection period. These uptake levels were significantly higher than the wider young adult population at that time *(p* < 0.0001 as compared to Leicester 18–24-year-olds; Appendix A), suggesting that these students were more proactive about accessing COVID-19 vaccines than their peers. High uptake may be partially attributable to a bias for pro-vaccine students to participate in the study and/or to surge vaccinations in the Leicester COVID-19 hotspot just prior to initiation of the survey. Our observation of a high willingness for uptake of COVID-19 vaccines (95%) was similar to the rates reported in an ONS study of UK university students [42] and may reflect the effectiveness of vaccine delivery and information campaigns targeted to students. Intriguingly, 93% (37/40) of the COVID-19 vaccine-hesitant individuals reported having had at least one of the MMR and MenACWY vaccines, suggesting that these individuals are either specifically concerned about the COVID-19 vaccines or that vaccine hesitancy has developed during their transition to adulthood.

### 4.1. Determinants of Vaccine Hesitancy among University Students

Identifying vaccine-hesitant individuals is a key concern for vaccination programmes. In our study, uptake of the COVID-19 vaccine or intention to vaccinate was found to be relatively high concurring with findings from other studies conducted with student groups. For example, Di Giuseppe et al. [19] performed a study among university students and employees in an Italian university in 2020 and reported that the willingness to obtain a COVID-19 vaccine was 84.1%. Similarly, research conducted among university students in the UK has also reported high vaccine uptake among this group, with uptake increasing over time [43,44].

Although vaccine hesitancy was low, we found that hesitancy was strongly linked to ethnicity and more specifically to Black ethnicity in our univariate and multivariate analyses, respectively. Furthermore, analysis of VAX scores for all individuals showed that Asian and Black ethnic groups had significantly lower VAX scores indicating a general trend towards hesitancy among the minority ethnic groups (Figure 1). Other studies have also found evidence of vaccine hesitancy associated with ethnicity [45,46,47,48,49] and specifically with students of Black ethnicity [48]. Hesitancy in these groups has been linked to discrimination, mistrust of healthcare organisations, misinformation, lower perceived vaccine efficacy/safety [45,50]. A substantial proportion of vaccine-hesitant individuals (37.5%; 15/40) in our study agreed with a statement that COVID-19 vaccines had not been thoroughly tested in different ethnic groups (Appendix A) suggestive of the element of mistrust that was previously shown to significantly influence vaccine-uptake decision-making [51].

A novel finding was of an association between vaccine hesitancy and students who lived at home and significantly lower average VAX scores for students living at home as compared to those living in private accommodation or other accommodation types (Figure 1). While the home student group was small (i.e., 7% of national and 19% of international students), higher vaccine hesitancy among these students may be due to these students being less concerned about spreading COVID-19 than those living away from home (50% and 62%, respectively). Conversely, the multivariate regression showed lower levels of vaccine hesitancy among students concerned about spreading COVID-19 (OR 0.5, 95% CI 0.3–0.81, *p* = 0.024) as also observed by Szmyd et al. [52] for a cohort of Polish students. This attitude of vaccine hesitancy among home students may have arisen as a result of reduced day-to-day social interactions leading to a lower perceived risk of the potential for spreading COVID-19.

Our study was performed a few months after concerns about side-effects of the AstraZeneca vaccine were widely publicised. Associations between these concerns and vaccine hesitancy were detected in both our univariate and multivariate models, indicating that this important factor could reduce vaccine uptake. Nevertheless, uptake and willingness levels were high, suggesting that other factors may override the perceived risks of side-effects.

### 4.2. MMR and MenACWY Vaccine Status and On-Campus Vaccination Preferences

An important strategy for increasing MMR and MenACWY vaccine uptake among young adults is making them aware of their vaccination status [53]. This is now demonstrably possible via digital applications such as the NHS App and EU Digital COVID certificate. Our survey found high levels of uncertainty among students about their MMR (11% did not know) and MenACWY (26% did not know) vaccine status and 3–4% who reported no uptake (Table 3). In a 2019/2020 questionnaire performed just prior to the pandemic, 16% and 54% of University of Leicester students reported not knowing their MMR and MenACWY vaccine status, respectively [31]. Estimates of actual vaccine uptake in England indicate that uptake is <90% for both the MMR and MenACWY vaccines (last reported in July 2019 and 2017/2018, respectively) with the MMR vaccination levels being below the >95% coverage recommended by the World Health Organisation for preventing measles and mumps outbreaks [28,30,54]. Many students who are uncertain about their vaccine status may not have had these vaccines, particularly international students (who reported high rates of uncertainty about their MMR and MenACWY vaccine status). These students will be at a higher potential risk of contracting and/or spreading the diseases targeted by these vaccines. Public Health England recommends that anyone who is uncertain about their vaccine status or has missed a vaccine dose should be offered these vaccines [55]. Most students will be unaware of this recommendation; indeed, 20% of UoL students did not know that these vaccines are free (Table 3)). High proportions of UK students and UK-based international were in favour of on-campus provision of MenACWY, MMR and COVID-19 vaccines, with the latter group potentially reflecting difficulties in understanding how to access the UK medical system. The statistically significant evidence of support for provision of on-campus COVID-19 vaccinations indicates that students value easy access and that this strategy could help to address deficits in vaccine uptake of all vaccines relevant to this age group.

### 4.3. Recommendations

Harnessing new approaches developed during the UK COVID-19 vaccine roll-out is a potential positive legacy of the pandemic to build back better for future generations. Empowering digitally aware young people to take responsibility for their own health and to engage in community health policies is an achievable, cost-efficient outcome with far-reaching personal and population benefits. A key recommendation is for provision of vaccine status and access information for all vaccines on digital platforms (e.g., NHSapp in the UK) so that individuals can make informed decisions about taking up missed vaccinations. Specific delivery of vaccines to international students should be a gold standard for the university sector combined with wide adoption of on-campus vaccination programmes. The benefits of implementation of these recommendations would be protection of more individuals and improved population protection through reduced transmission.

### 4.4. Strengths and Limitations

A strength of this study is that it is the first to simultaneously evaluate uptake, knowledge and attitudes to COVID-19, MMR and MenACWY vaccines among university students. A further strength is the high number and ethnic diversity of the participant population. The use of multivariable regression was a strength that allowed for adjustment for confounders and for identification of significant associations between variables and vaccination parameters with the potential to inform vaccination policies.

Findings from this study may, however, be affected by the inherent limitations of cross-sectional studies. Our approach of utilising emails to send out invitations and an online survey form may have limited access for some potential participants. The self-selecting nature of the response cohort and a response rate of 8%, despite incentives, indicates that there may be bias due to demographics. The strengths and limitations arising from a range of demographics were considered above, but it is possible that biases from other unaccounted demographics (e.g., socioeconomic status) may confound generalisability of our data to the wider UK student population. We note that distribution in ethnic group and gender differ in our response cohort to both the University of Leicester and the wider UK student demographic and have attempted to mitigate these effects by using a weighted multivariate analysis. A further limitation is that a pilot study was not completed with the entire questionnaire. A significant potential limitation of our study, and inherent in many studies of vaccination, is enhanced participation by individuals with pro-vaccine attitudes and reduced participation by vaccine-hesitant individuals. Our study may also have been subject to social desirability bias due to the study survey being delivered through the University of Leicester email system and being promoted by the senior university team. Our fully anonymised survey system and online formats was designed to counter both of these biases while inducements to participate was designed to minimise the former bias. Our level of vaccine hesitancy as determined by vaccine uptake is similar to other studies. However, this strict determination of vaccine hesitancy may have missed the full range of hesitancy and excluded students who obtained the vaccine despite having a degree of vaccine hesitancy.

## 5. Conclusions

The findings from this study indicate that there may be differences in uptake and access to the COVID-19, MenACWY and MMR vaccines among university students. Students of Black ethnicity and those residing at home were less likely to be vaccinated with COVID-19 vaccines. Further research on the reasons for hesitancy may be required in order to delivery more effective, ‘tailored’ vaccine information and to develop methods for enhancing trust and acceptance of vaccines in these groups. High levels of uncertainty about personal vaccine status and availability of the MMR and MenACWY vaccines were observed and are likely to impact vaccine uptake. On campus vaccination delivery was found to be widely favoured particularly by on-campus and international students. These knowledge gaps and delivery approaches should be considered in future student-focussed vaccination campaigns and explored through additional research. Our findings indicate that adopting ‘best practices’ of easy access and digital vaccine information within the university-sector may break down barriers and future-proof uptake of all required vaccines among students.

## Figures and Tables

**Figure 1 vaccines-10-01287-f001:**
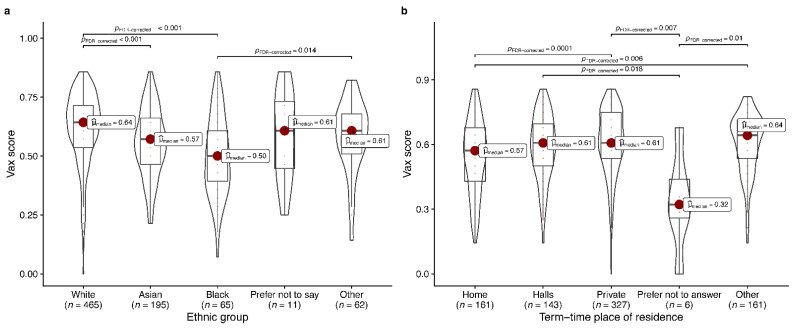
VAX scores for a range of ethnic groups and study residences. The VAX score was determined for each student from responses to four statements about vaccines (statements 17.1, 17.2, 17.3 and 17.5 in Question 17; see Appendix A and Appendix A). VAX scores range from 0 to 1 representing high to low vaccine hesitancy. The VAX scores were determined for all individuals in four broad ethnic groups (**a**) or places of residence during the university term (**b**). Graphs show violin plots with the median scores indicated by a red circle. Box, IQR; line, IQR + 1.5 times IQR; line within box, median. *p*-values were derived using pairwise Dunn tests with FDR correction.

**Table 1 vaccines-10-01287-t001:** Demographic characteristics of study participants and results of an unweighted univariate analysis of vaccine hesitancy.

Characteristic	All Respondents (*n* = 827) ^1^	Hesitancy Group			
		Willing (*n* = 721) ^1^	Hesitant (*n* = 40) ^1^	*p*-Value ^2^	q-Value ^3^
Ethnic group				**<0.001**	**<0.001**
White	479 (58%)	428 (59%)	16 (40%)		
Asian	203 (25%)	180 (25%)	6 (15%)		
Black	69 (8.3%)	48 (6.7%)	12 (30%)		
Prefer not to say	11 (1.3%)	7 (1.0%)	3 (7.5%)		
Other	65 (7.9%)	58 (8.0%)	3 (7.5%)		
Gender				0.4	0.4
Female	548 (66%)	477 (66%)	25 (62%)		
Male	255 (31%)	222 (31%)	14 (35%)		
Other	18 (2.2%)	17 (2.4%)	0 (0%)		
Course studied				**0.026**	0.060
Humanities, Law and Social Science	349 (42%)	288 (40%)	23 (57%)		
Natural and Life Sciences	302 (37%)	261 (36%)	14 (35%)		
Medicine and allied	176 (21%)	172 (24%)	3 (7.5%)		
Home/international student				0.13	0.2
UK student	734 (90%)	643 (90%)	31 (84%)		
UK-based international student	48 (5.9%)	41 (5.7%)	5 (14%)		
Non-UK international student	38 (4.6%)	34 (4.7%)	1 (2.7%)		
Non-term residence				0.10	0.14
Local	160 (21%)	135 (20%)	10 (30%)		
National	534 (68%)	473 (69%)	17 (52%)		
International	86 (11%)	75 (11%)	6 (18%)		
Age group				**0.030**	0.060
22+	297 (36%)	266 (37%)	8 (20%)		
<= 21	529 (64%)	454 (63%)	32 (80%)		
Experience of harassment				0.8	0.8
No	752 (92%)	659 (93%)	35 (92%)		
Yes	62 (7.6%)	53 (7.4%)	3 (7.9%)		
COVID-19-related death in known contact				0.5	0.5
No	475 (58%)	418 (58%)	21 (52%)		
Yes	346 (42%)	299 (42%)	19 (48%)		
Concern of vaccine side-effects (Oxford/AstraZeneca)				**<0.001**	**<0.001**
1 (Strongly disagree)	171 (21%)	154 (22%)	4 (10%)		
2	153 (19%)	141 (20%)	1 (2.6%)		
3	99 (12%)	91 (13%)	2 (5.1%)		
4	105 (13%)	88 (13%)	8 (21%)		
5	104 (13%)	92 (13%)	3 (7.7%)		
6	75 (9.4%)	65 (9.3%)	4 (10%)		
7 (Strongly agree)	94 (12%)	67 (9.6%)	17 (44%)		
Concern over hospitalisation				**0.038**	0.068
0	392 (48%)	337 (47%)	25 (66%)		
1	339 (41%)	305 (42%)	8 (21%)		
2	60 (7.3%)	50 (6.9%)	3 (7.9%)		
3	33 (4.0%)	28 (3.9%)	2 (5.3%)		
Concern over spreading COVID-19 to others				**0.003**	**0.007**
0	82 (10.0%)	63 (8.8%)	11 (29%)		
1	226 (28%)	203 (28%)	10 (26%)		
2	297 (36%)	264 (37%)	12 (32%)		
3	216 (26%)	189 (26%)	5 (13%)		
VAX score				**<0.001**	**<0.001**
	0.61 (0.50, 0.71)	0.61 (0.50, 0.71)	0.39 (0.25, 0.53)		
Unknown ^4^	29	19	6		
Year of study				0.050	0.081
Foundation	16 (1.9%)	15 (2.1%)	1 (2.5%)		
1	269 (33%)	228 (32%)	18 (45%)		
2	217 (26%)	188 (26%)	11 (28%)		
3	234 (28%)	201 (28%)	10 (25%)		
4	91 (11%)	89 (12%)	0 (0%)		
Term-time residence				**<0.001**	**<0.001**
Home	171 (21%)	134 (19%)	14 (35%)		
Halls	152 (18%)	137 (19%)	8 (20%)		
Private	333 (40%)	301 (42%)	10 (25%)		
Prefer not to answer	6 (0.7%)	2 (0.3%)	4 (10%)		
Other	165 (20%)	147 (20%)	4 (10%)		
Self-determination score	79 (71, 88)	80 (72, 88)	76 (68, 84)	0.3	0.4
Unknown ^4^	86	71	5		

^1^ 0, vaccine-willing; 1, vaccine-hesitant; N, number of participants (%); median (IQR); ^2^ Fisher’s exact test; Pearson’s chi-squared test; Wilcoxon rank sum test; ^3^ false discovery rate correction for multiple testing; ^4^ unknown, number participants with incomplete answers (continuous variables). Numbers for some (categorical) characteristics do not add up to the total survey population (*n* = 827) due to missing values. Significant *p*-values (<0.05) are highlighted in bold.

**Table 2 vaccines-10-01287-t002:** Multivariate analysis of vaccine hesitancy.

		Hesitancy (Unweighted)				Hesitancy (Weighted)				
Characteristic	N	OR ^1^	95% CI ^1^	*p*-Value	q-Value ^2^	N	OR ^1^	95% CI ^1^	*p*-Value	q-Value ^2^
Gender										
Female	419	—	—			299	—	—		
Male	193	0.84	0.22, 2.89	0.8	>0.9	147	0.63	0.21, 1.85	0.4	0.6
Unknown	5	0.00		>0.9	>0.9					
Ethnic group										
White	377	—	—			271	—	—		
Asian	145	0.23	0.04, 1.03	0.072	0.2	97	0.24	0.04, 1.52	0.13	0.2
Black	43	6.17	1.48, 26.7	**0.012**	0.082	33	7.01	1.81, 27.3	**0.005**	**0.021**
Other	52	1.40	0.17, 8.22	0.7	>0.9	39	1.37	0.12, 15.4	0.8	>0.9
Age group										
22+	230	—	—			169	—	—		
<= 21	387	2.92	0.78, 12.6	0.13	0.3	276	3.25	0.73, 14.4	0.12	0.2
Course studied ^3^										
H, Law, Soc	240	—	—			175	—	—		
Nat/Life Sci	229	1.04	0.31, 3.44	>0.9	>0.9	158	0.93	0.32, 2.70	0.9	>0.9
Med/Allied	148	0.16	0.02, 0.91	0.061	0.2	110	0.10	0.02, 0.50	**0.005**	**0.021**
Non-term residence										
Local	120	—	—			92	—	—		
National	429	0.61	0.13, 2.79	0.5	0.8	305	0.71	0.20, 2.46	0.6	0.7
International	68	1.19	0.19, 7.23	0.8	>0.9	49	1.76	0.44, 7.15	0.4	0.6
Experience of harassment										
No	570	—	—			413	—	—		
Yes	47	2.42	0.34, 13.1	0.3	0.5	31	3.61	0.51, 25.8	0.2	0.3
COVID-19-related death in known contact										
No	351	—	—			267	—	—		
Yes	266	5.05	1.65, 17.8	**0.007**	0.068	197	7.49	2.06, 27.2	**0.002**	**0.014**
Concern of vaccine side-effects (Oxford/AstraZeneca)										
	593	1.71	1.29, 2.37	**<0.001**	**0.010**	610	1.72	1.23, 2.39	**0.001**	**0.013**
Concern over hospitalisation										
	593	0.56	0.22, 1.25	0.2	0.3	610	0.68	0.24, 1.89	0.5	0.6
Concern over spreading COVID-19 to others										
	593	0.52	0.27, 0.96	**0.041**	0.14	610	0.45	0.25, 0.81	**0.008**	**0.024**
Year of study										
	593	0.81	0.39, 1.59	0.6	0.8	610	0.94	0.49, 1.81	0.9	>0.9
Term-time residence										
Home	124	—	—			93	—	—		
Halls	119	0.23	0.04, 1.24	0.093	0.2	96	0.20	0.04, 1.03	0.054	0.13
Private	257	0.17	0.03, 0.83	**0.032**	0.14	186	0.13	0.04, 0.38	**<0.001**	**0.004**
Other	117	0.09	0.00, 0.62	**0.036**	0.14	78	0.07	0.01, 0.67	**0.021**	0.057
Self-determination score										
	593	1.01	0.96, 1.06	0.7	0.9	610	1.00	0.96, 1.04	>0.9	>0.9

^1^ OR = odds ratio, CI = confidence interval; ^2^ false discovery rate correction for multiple testing. ^3^ Abbreviations: H, humanities; Soc, social sciences; Nat/Life Sci, natural and life sciences; Med/Allied, medicine and allied courses. Significant *p*-values (<0.05) are highlighted in bold.

**Table 3 vaccines-10-01287-t003:** Comparative knowledge and attitudes to on campus delivery of COVID-19, MMR and MenACWY vaccines.

Question	Vaccine Types	Possible Answers	UK Students (*N* = 734)	UK-Based International Students (*N* = 48)	Non-UK International Student (*N* = 38)	*p*-Value
Qu. 13. Have you received the following vaccines either as a child or adult	13.1.a. MMR (measles, mumps and rubella)	Yes	659 (90%)	28 (58%)	23 (61%)	<0.001
		Don’t Know	58 (7.9%)	19 (40%)	12 (32%)	
		No	17 (2.3%)	1 (2.1%)	3 (7.9%)	
	13.2.a. MenACWY (meningitis)	Yes	538 (73%)	17 (35%)	18 (47%)	<0.001
		Don’t Know	170 (23%)	25 (52%)	17 (45%)	
		No	26 (3.5%)	6 (12%)	3 (7.9%)	
Qu. 14. Are you aware that all of these vaccines are free in the UK for students?	14.1.a. MMR/MenACWY	Yes	623 (85%)	23 (48%)	15 (39%)	<0.001
		No	111 (15%)	25 (52%)	23 (61%)	
	14.2.b. COVID-19	Yes	693 (94%)	43 (90%)	23 (61%)	<0.001
		No	41 (5.6%)	5 (10%)	15 (39%)	
Qu. 15. If vaccines were offered on campus, would this affect your decision to be vaccinated?	15.1.a. MenACWY/MMR	Definitely increase	313 (43%)	25 (52%)	10 (26%)	n/a
		Somewhat increase	103 (14%)	8 (17%)	9 (24%)	
		Neither	290 (40%)	13 (27%)	18 (47%)	
		Somewhat decrease	2 (0.3%)	0 (0%)	0 (0%)	
		Definitely decrease	2 (0.3%)	0 (0%)	1 (2.6%)	
		Don’t know	24 (3.3%)	2 (4.2%)	0 (0%)	
	15.2.a. COVID-19	Definitely increase	331 (45%)	30 (62%)	11 (29%)	n/a
		Somewhat increase	116 (16%)	6 (12%)	9 (24%)	
		Neither	264 (36%)	10 (21%)	17 (45%)	
		Somewhat decrease	3 (0.4%)	2 (4.2%)	0 (0%)	
		Definitely decrease	2 (0.3%)	0 (0%)	1 (2.6%)	
		Don’t know	18 (2.5%)	0 (0%)	0 (0%)	

## Data Availability

Data are available from the UK Data Service at https://reshare.ukdataservice.ac.uk/855372/, accessed on 9 August 2022 (10.5255/UKDA-SN-855372).

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
