# Peer review of "Cross-Sectional Study of University Students’ Attitudes to ‘On Campus’ Delivery of COVID-19, MenACWY and MMR Vaccines and Future-Proofing Vaccine Roll-Out Strategies"

_vaccines, 2022, doi:10.3390/vaccines10081287_

Round 1
Reviewer 1 Report
This is a timely study, clearly analysed and well-reported. The main weakness is the low response rate to the survey, which is acknowledged by the authors.
I have no substantive comments to make on improving the manuscript.
Author Response
We appreciate this reviewers positive comments on our manuscript and note that there are no points to address
Reviewer 2 Report
This is a concise and well-written study on vaccine hesitancy among UK students. The main problem in the study is the low (about 7%) response rate of the students, but the authors describe this issue by themselves in the section 4.4. and consider its importance. I find the study acceptable for publication in Vaccines.
Author Response
We thank the reviewer for their positive comments on our manuscript and note that their are no comments to address
Reviewer 3 Report
This is an interesting and timely manuscript on vaccine hesitancy. Authors started off well, however the manuscript needs some updating before it can be considered for final publication. The authors need to review extant literature on vaccine hesitancy. They all need to better connect their findings to existing studies. Some relevant studies in the domain are:
Chi et al., (2022). Evolving effects of COVID-19 safety precaution expectations, risk avoidance, and socio-demographics factors on customer hesitation toward patronizing restaurants and hotels. Journal of Hospitality Marketing & Management, 1-17.
Ramkissoon, H. (2021). Social Bonding and Public Trust/Distrust in COVID-19 Vaccines. Sustainability, 13(18), 10248.
The methodology is ok. The conclusion ties well to the rest of the paper. Thank you.
Author Response
We thank the reviewer for their positive comments on our paper and will work to address the deficiencies in our comparisons to extant literature.
The authors need to review extant literature on vaccine hesitancy. They all need to better connect their findings to existing studies. Some relevant studies in the domain are: Chi et al., (2022). Evolving effects of COVID-19 safety precaution expectations, risk avoidance, and socio-demographics factors on customer hesitation toward patronizing restaurants and hotels. Journal of Hospitality Marketing & Management, 1-17. Ramkissoon, H. (2021). Social Bonding and Public Trust/Distrust in COVID-19 Vaccines. Sustainability, 13(18), 10248.
Response: We thank the reviewer for these suggestions. We have made fresh searches within the PubMed database to identify recent and relevant information on vaccine hesitancy. We have included some of the most relevant articles and findings in the Introduction and Discussion sections of our manuscript (see lines 53-63 and 335-368). We have also reviewed the two suggested references and included Ramkissoon (2021) due to its relevance to our study but not Chi et al (2022) as this was outside the scope of our work.
The relevant sections and new references are copied here for ease of reference
Common demographics for COVID-19 vaccine hesitancy in the reported literature include females, younger age groups, being from a minority ethnic group and lower education or income levels [12-14]. Studies of student populations have yielded a range of findings. Factors associated with higher vaccine acceptance are knowledge of COVID-19 vaccines, trust in authorities and high perceived vaccine safety and effectiveness and uptake of vaccination in the family [10, 11, 15-20-]. Consequently, perceived accessibility barriers (physical or financial), concerns about vaccine side-effects, speed of development and previous COVID-19 infection have been associated with vaccine hesitancy [11, 18, 20-23]. A potentially important issue is whether news about the rare but serious side-effect of blood clots associated with the licensed AstraZeneca COVID-19 vaccine might have affected uptake among students [24, 25].4.1. Determinants of vaccine hesitancy among university students
Identifying vaccine hesitant individuals is a key concern for vaccination programmes. In our study, uptake of the COVID-19 vaccine or intention to vaccinate was found to be relatively high concurring with findings from other studies conducted with student groups. For example, Giuseppe et al. [19] performed a study among university students and employees in an Italian university in 2020 and reported that the willingness to obtain a COVID-19 vaccine was 84.1%. Similarly, research conducted among university students in the UK have also reported high vaccine uptake among this group, with uptake increasing over time [43, 44].
Although vaccine hesitancy was low, we found that hesitancy was strongly linked to ethnicity and more specifically to black ethnicity in our univariate and multivariate analyses, respectively. Furthermore analysis of VAX scores for all individuals showed that Asian and Black ethnic groups had significantly lower VAX scores indicating a general trend towards hesitancy among the minority ethnic groups (Figure 1). Other studies have also found evidence of vaccine hesitancy associated with ethnicity [45-49,] and specifically with students of black ethnicity [48]. Hesitancy in these groups has been linked to discrimination, mistrust of healthcare organisations, misinformation, lower perceived vaccine efficacy/safety [45, 50,]. A substantial proportion of vaccine hesitant individuals (37.5%; 15/40) in our study agreed with a statement that COVID-19 vaccines had not been thoroughly tested in different ethnic groups (S9 Table) suggestive of the element of mistrust that was previously shown to significantly influence vaccine-uptake decision-making [51].
New references
16. Jiang N, Wei B, Lin H, Wang Y, Chai S, Liu W. Nursing students' attitudes, knowledge and willingness of to receive the coronavirus disease vaccine: A cross-sectional study. Nurse Educ Pract., 2021, 55:103148. doi: 10.1016/j.nepr.2021.103148. 19. Di Giuseppe, G.; Pelullo, C.P.; Della Polla, G.; Pavia, M.; Angelillo, I.F. Exploring the Willingness to Accept SARS-CoV-2 Vaccine in a University Population in Southern Italy, September to November 2020. Vaccines2021, 9, 275. https://doi.org/10.3390/vaccines9030275 20. Jiang N, Gu P, Liu K, Song N & Jiang X. Acceptance of COVID-19 vaccines among college students: a study of the attitudes, knowledge, and willingness of students to vaccinate. Human Vaccines & Immunotherapeutics, 2021, 17:12, 4914-4924, DOI: 1080/21645515.2021.201307721. Lucia VC, Kelekar A, Afonso NM. COVID-19 vaccine hesitancy among medical students. J Public Health (Oxf). 2021 Sep 22;43(3):445-449. doi: 10.1093/pubmed/fdaa230
22. Tavolacci MP, Dechelotte P, Ladner J. COVID-19 Vaccine Acceptance, Hesitancy, and Resistancy among University Students in France. Vaccines (Basel). 2021 Jun 15;9(6):654. doi: 10.3390/vaccines9060654
23. Riad A, Pokorná A, Antalová N, Krobot M, Zviadadze N, Serdiuk I, Koščík M, Klugar M. Prevalence and Drivers of COVID-19 Vaccine Hesitancy among Czech University Students: National Cross-Sectional Study. Vaccines (Basel). 2021 Aug 25;9(9):948. doi: 10.3390/vaccines9090948
49. Nguyen, L.H., Joshi, A.D., Drew, D.A. et al.Self-reported COVID-19 vaccine hesitancy and uptake among participants from different racial and ethnic groups in the United States and United Kingdom. Nat Commun 13, 636 (2022). https://doi.org/10.1038/s41467-022-28200-3
50. Razai M S, Osama T, McKechnie D G J, Majeed A. Covid-19 vaccine hesitancy among ethnic minority groups BMJ 2021; 372 :n513 doi:10.1136/bmj.n513
51. Ramkissoon, H. Social Bonding and Public Trust/Distrust in COVID-19 Vaccines. Sustainability2021, 13, 10248. https://doi.org/10.3390/su131810248
52. Szmyd B, Bartoszek A, Karuga FF, Staniecka K, Błaszczyk M, Radek M. Medical Students and SARS-CoV-2 Vaccination: Attitude and Behaviors. Vaccines(Basel). 2021 Feb 5;9(2):128. doi: 10.3390/vaccines9020128
Reviewer 4 Report
Methods
a) The Authors should clarify how has been determined the sample size.
b) It is not given any information whether participants were not able to continue to the next question of the questionnaire if they failed to provide a response to an item.
c) It is not given any information regarding a pilot study for testing the survey questionnaire.
d) The Authors should describe the survey questionnaire items in more details.
e) No information is given regarding how the Informed consent has been obtained (written or oral) from respondents who participated in the study.
f) The Authors should clarify about the face-validity testing of the questions with an explanation of the validity of the content of the questions with regard to the research aims. The Authors should clarify how they had estimated the reliability, or internal consistency, of the questions by using, for example the Cronbach’s alpha in order to measure whether or not a score is reliable.
g) In the statistical analysis it is particularly relevant to describe the multivariate logistic regression model(s) developed and the outcome(s), the variables included and the rationale why they are included, and the model building strategy. Moreover, it should be indicated the p-value that has been used to determine statistical significance and if the tests were one-side or two-sided.
Results
a) No information is given about those who refused to participate. Was there any attempt to quantify the response bias: information about non-responders. It would be useful to have some kind of indication of comparability with non-respondents. Is there any population-based data available? How did they differ from those in the sample, how representative is the sample and were the findings representative of UK?
Discussion
a) There is a lack of comparison with the results of recent studies conducted in other geographic areas. The work should therefore be enriched in such a way as to become self-supporting by photographing the context and what is around it. For example, the following articles should be cited Di Giuseppe et al. Vaccines (Basel) 2021;9:275; Jiang et al. Hum Vaccin Immunother 2021;17(12):4914-24; Szmyd et al. Vaccines (Basel) 2021;9:128.
b) The pivotal role of healthcare providers as source of information with a positive impact towards vaccination attitudes, knowledge, and willingness should be stressed and studies supporting this statement should be added. For example, the following articles should be cited Di Giuseppe et al. Expert Rev Vaccines 2021;20:881-9; Wang et al. Vaccines (Basel) 2021;9:29.
c) The paragraph regarding the limitations of the study should discuss all limits such as, for example, the study design, the method of sampling, the representativeness of the sample, and the social desirability bias.
References
a) The manuscript is not well referenced. The References list is not updated, since several articles conducted in different countries and published on peer-reviewed journals have been not included.
Tables
a) In Table 1 do not include the number of those who did not answer to the question (Unknown). It should be clarified that the number of respondents is not the same for all variables. Add a footnote Numbers for some characteristics do not add up to the total number of the study population due to missing values.
Author Response
Reviewer 4
Methods
- The Authors should clarify how has been determined the sample size.
Response: We have added the following text into the Methods section to answer this question: Sample size was determined by the total number of surveys completed with all responses being considered in the analysis.
2. It is not given any information whether participants were not able to continue to the next question of the questionnaire if they failed to provide a response to an item.
Response: Every question, except 28., were compulsory and the questionnaire could not be submitted if answers were not provided to these compulsory questions. Most of the compulsory questions, included a “Prefer not to answer” or “Don’t know” response giving participants the possibility of not providing specific answers. The exceptions were 4, 9, 14, 16 and 29; these questions were either Open Text boxes or required YES/NO responses to specific statements.
We have modified the Methods section to include the following sentence: Most of the questions were compulsory, but included either ‘Prefer not to answer’ or ‘Don’t know’ responses (Table S1).
We have also modified the Table S1 to indicate the question that was not compulsory.
3. It is not given any information regarding a pilot study for testing the survey questionnaire.
Response: This information is already provided in Section 2.3 where we clearly set out the sources of questions. In this section we note that 13 of the questions were previously trialled in other studies whilst the other questions were written for this study.
In response to the reviewer, we note that a pilot study was not possible due to the constraints of our funding programme (a rapid and short-term COVID-19 response scheme), the academic timetable and the rapid rollout of the vaccination programme.
4. The Authors should describe the survey questionnaire items in more details.
Response: The full text of the survey questions are provided in Table S1 in Supplementary Information. We did not think it was appropriate to included the full text of questions in the main manuscript as this would unnecessarily lengthen the text.
5. No information is given regarding how the Informed consent has been obtained (written or oral) from respondents who participated in the study.
Response: This information was provided in section 2.3, lines 120-122. We have slightly modified this sentence to make this clearer.
6. The Authors should clarify about the face-validity testing of the questions with an explanation of the validity of the content of the questions with regard to the research aims. The Authors should clarify how they had estimated the reliability, or internal consistency, of the questions by using, for example the Cronbach’s alpha in order to measure whether or not a score is reliable.
Response: We thank the reviewer for this comment. We have performed some additional analyses to address the internal consistency of our findings and have modified the text accordingly. The new analyses include both calculation of a Cronbach’s alpha value and addition of an extra test. The new test uses a more powerful (non-parametric) test and shows that the vax score ouputs correlate with the direct measure of vaccine hesitancy. These changes have only altered the results very slightly and are reflected by inclusion of new text (see below) and a revised version of figure 1.
New text: The Asian ethnic group had significantly lower VAX scores than both White and Other ethnic groups, while the Black ethnic group had significantly lower VAX scores than White ethnicity, indicating a higher level of vaccine hesitancy in Asian and Black groups (Figure 1). Similarly, we observed that home students had significantly lower VAX scores than students living in private or other accommodation (Figure 1). The mean VAX score for students living in halls was higher but not significantly different to those living at home, indicating a trend for home students to be more vaccine hesitant than those who lived in other locations during this academic year. VAX scores in our sample had a low but acceptable internal consistency score (Cronbach’s alpha = 0.62 (95%CI 0.58-0.66)), and a negative association with our independent measure of vaccine hesitancy (rank biserial correlation r = 0.63; Wilcoxon rank sum test P < 0.0001), as expected.
7. In the statistical analysis it is particularly relevant to describe the multivariate logistic regression model(s) developed and the outcome(s), the variables included and the rationale why they are included, and the model building strategy. Moreover, it should be indicated the p-value that has been used to determine statistical significance and if the tests were one-side or two-sided.
Response: The components of the multivariate logistic regression model are described in the some depth in section 2.5. We modified this section to indicate that the focus of our model was to identify significant factors affecting vaccine hesitancy rather than to produce a predictive model of vaccine hesitancy. We have also included the p-value utilised in these models. The relevant modified paragraphs are now as follows:-
Multivariable analysis was performed using logistic regression (glm; binomial link function) on both unweighted and weighted survey results, with vaccine hesitancy and preference for on-campus vaccinations (COVID-19 and MMR) as dependent variables. Predictors included gender, ethnic group, age group, course studied, year of study, experience of harassment, experience of COVID-19 related death, concern over side-effects from the Oxford/AstraZeneca vaccine, concern over hospitalization from COVID-19, concern over spreading COVID-19, home area (local, national, international), residence while studying (home, halls of residence, private accommodation, other) and a psychometric score on self-determination/fatalism. Home area was derived using information on post-codes and international status with students being classified as local if they came from either Leicester or the wider county of Leicestershire, ‘national’ for students from the rest of the UK and ‘international’ for students ordinarily living overseas. Experiences of COVID-19 related deaths were classified into a Yes or No category according to responses to question 21 (Table S1) with the Yes responses including family member(s), friend(s) or someone else. Survey data were weighted using the raking method in the survey package version 4.0 for R based on national student distributions for ethnic group (White, Asian, Black and Other) and gender (Tables S2, S3 and S4). Constraints in the national data made it necessary to remove students of unknown gender (n = 6). Given the low rate of vaccine hesitancy and the inherent limitations on creating meaningful and representative training and test data subsets, no attempt was made to assess model performance using e.g. ROC AUC. Multivariate models were used to identify significant factors influencing vaccine hesitancy rather than for the purpose of generating an effective predictive model of hesitancy.
Statistical differences between the distributions of VAX scores for different groups were determined using pairwise Dunn tests with FDR correction.
All tests were two-sided with a corrected p-value (FDR correction) of < 0.05 considered significant.
Results
- No information is given about those who refused to participate. Was there any attempt to quantify the response bias: information about non-responders. It would be useful to have some kind of indication of comparability with non-respondents. Is there any population-based data available? How did they differ from those in the sample, how representative is the sample and were the findings representative of UK?
Response: It is not completely clear if the reviewer is concerned about the representativeness of our study population in general or representativeness of the vaccine hesitant population per se.
To address the first point, we compared the ethnicity, gender and year group demographics of our survey respondents to both the University of Leicester and UK Universities demographics in the supplementary information (see Tables S3, S4 and S5) and the first paragraph of the results. These data show that our study population is, with some caveats, representation of UK university student populations and hence will be an excellent benchmark and comparitor for other studies.
With regard to vaccine uptake and vaccine hesitancy, we discuss this at some length in the Discussion (lines 321-334) with the limitations being acknowledged in section 4.4. We indicated that our vaccine hesitancy levels were similar to those found in the ONS study of UK university students. Our study appears therefore to be in line with similar studies with regard to sampling of the vaccine hesitancy population.
Overall, we have provided a significant amount of information with regard to the representativeness of the relevant study population in general and of vaccine hesitant individuals in this population specifically. We have also acknowledged the limitations with regard to sampling vaccine hesitant individuals, which is a problem common to all surveys in this area of research. We have attempted to combat this issue by utilizing the VAX scores to identify vaccine hesitancy by an alternative measure and shown the utility of this approach.
We hope this summary has answered and assuaged these concerns of this reviewer.
Discussion
1. There is a lack of comparison with the results of recent studies conducted in other geographic areas. The work should therefore be enriched in such a way as to become self-supporting by photographing the context and what is around it. For example, the following articles should be cited Di Giuseppe et al. Vaccines (Basel) 2021;9:275; Jiang et al. Hum Vaccin Immunother 2021;17(12):4914-24; Szmyd et al. Vaccines (Basel) 2021;9:128.
Response: We have revised our Discussion section to make further connections between our results and the existing literature. We have specifically compared our results to the studies of Di Guiseppe et al. and Szmyd et al. as these studies covered similar study populations in other countries. A comprehensive comparison with the published literature is however beyond the scope of this manuscript given the restricted word count and the differing aims and methodologies of other studies and the differing vaccine delivery programmes across the. world. We would also like to point out that some of our findings, for example, the higher odds of hesitancy among students staying at home, are quite novel and therefore difficult to situate within existing knowledge.
2. The pivotal role of healthcare providers as source of information with a positive impact towards vaccination attitudes, knowledge, and willingnessshould be stressed and studies supporting this statement should be added. For example, the following articles should be cited Di Giuseppe et al. Expert Rev Vaccines 2021;20:881-9; Wang et al. Vaccines (Basel) 2021;9:29.
Response: We acknowledge that this is an important area and indeed that we have collected information relevant to this topic. We will be looking to evaluate this information in future manuscripts but feel that this area is beyond the scope of the current manuscript.
3. The paragraph regarding the limitations of the study should discuss all limits such as, for example, the study design, the method of sampling, the representativeness of the sample, and the social desirability bias.
Response; We have modified this paragraph to include some additional potential limitations and our approach to mitigate demographic short-comings. The relevant paragraph now reads:-
Findings from this study may, however, be affected by the inherent limitations of cross-sectional studies. Our approach of utilizing emails to send out invitations and an online survey form may have limited access by some potential participants. The self-selecting nature of the response cohort and a response rate of 8%, despite incentives, indicates that there may be bias due to demographics. The strengths and limitations arising from a range of demographics were considered above but it is possible that biases from other unaccounted demographics (e.g. socioeconomic status) may confound generalisability of our data to the wider UK student population. We note that distribution in ethnic group and gender differ in our response cohort to both the University of Leicester and the wider UK student demographic and have attempted to mitigate these effects by using a weighted multivariate analysis. A significant potential limitation of our study, and inherent in many studies of vaccination, is enhanced participation by individuals with pro-vaccine attitudes and reduced participation by vaccine hesitant individuals. Our fully anonymised survey system and inducements to participate was designed to minimise this bias. Our level of vaccine hesitancy as determined by vaccine uptake is similar to other studies. However, this strict determination of vaccine hesitancy may have missed the full range of hesitancy and excluded students who obtained the vaccine despite having a degree of vaccine hesitancy.
References
- The manuscript is not well referenced. The References list is not updated, since several articles conducted in different countries and published on peer-reviewed journals have been not included.
Response: We thank the Reviewer for this comment and have revised our Reference list to include recent, relevant studies.
Tables
- a) In Table 1 do not include the number of those who did not answer to the question (Unknown). It should be clarified that the number of respondents is not the same for all variables. Add a footnote Numbers for some characteristics do not add up to the total number of the study population due to missing values.
Response: we have removed these numbers for the categorical variables but not for the continuous variables as it is not possible to determine the % missingness from just mean and standard deviation values. We hope that this is a reasonable compromise.
Round 2
Reviewer 3 Report
This is a much improved version, thank you. I recommend acceptance of your paper.
Reviewer 4 Report
Methods
a) Again, the Authors should clarify how has been determined the sample size.
b) The lack of a pilot study for testing the survey questionnaire is an important limitation that should be acknowledged.
c) Again, in the statistical analysis it is particularly relevant to describe the multivariate logistic regression model(s) developed and the outcome(s), the variables included and the rationale why they are included, and the model building strategy.
Results
a) Again, the question is clear since no information is given about those who refused to participate. Was there any attempt to quantify the response bias: information about non-responders. It would be useful to have some kind of indication of comparability with non-respondents. Is there any population-based data available? How did they differ from those in the sample, how representative is the sample and were the findings representative of UK?
Discussion
b) Again, the pivotal role of healthcare providers as source of information with a positive impact towards vaccination attitudes, knowledge, and willingness should be stressed and studies supporting this statement should be added. For example, the following articles should be cited Di Giuseppe et al. Expert Rev Vaccines 2021;20:881-9; Wang et al. Vaccines (Basel) 2021;9:29.
c) Again, the paragraph regarding the limitations of the study should discuss all limits such as, for example, the study design and the social desirability bias.
Tables
a) Again in Table 1 do not include the number of those who did not answer to the question (Unknown) and it is not clear at all the meaning that for the continuous variables as it is not possible to determine the % missingness from just mean and standard deviation values.
Author Response
Thank you for your comments, please find our reply in the attachment.
